# Microstructure and Texture Evolution during Superplastic Deformation of SP700 Titanium Alloy

**DOI:** 10.3390/ma15051808

**Published:** 2022-02-28

**Authors:** Ning Tian, Wenjun Ye, Xiaoyun Song, Songxiao Hui

**Affiliations:** 1State Key Laboratory of Nonferrous Metals and Processes, GRINM Group Co., Ltd., Beijing 100088, China; tianningnene@163.com (N.T.); huisx@grinm.com (S.H.); 2GRIMAT Engineering Institute Co., Ltd., Beijing 101407, China; 3General Research Institute for Nonferrous Metals, Beijing 100088, China

**Keywords:** SP700 titanium alloy, superplasticity, microstructure, texture, deformation mechanism

## Abstract

The superplastic tensile test was carried out on SP700 (Ti-4.5Al-3V-2Mo-2Fe) titanium alloy sheet at 760 °C by the method of maximum m value, and the microstructure characteristics were investigated to understand the deformation mechanism. The results indicated that the examined alloy showed an extremely fine grain size of ~1.3 μm and an excellent superplasticity with fracture elongation of up to 3000%. The grain size and the volume fraction of the β phase increased as the strain increased, accompanied by the elements’ diffusion. The β-stabilizing elements (Mo, Fe, and V) were mainly dissolved within the β phase and diffused from α to β phase furthermore during deformation. The increase in strain leads to the accumulation of dislocations, which results in the increase in the proportion of low angle grain boundaries by 15%. As the deformation process, the crystal of α grains rotated, and the texture changed, accompanied by the accumulation of dislocations. The phase boundary (α/β) sliding accommodated by dislocation slip was the predominant mechanism for SP700 alloy during superplastic deformation.

## 1. Introduction

Nowadays, titanium alloys have been widely applied in aerospace, biomedical, and industrial fields due to their excellent comprehensive properties, such as high specific strength and corrosion resistance [1,2,3,4,5,6]. However, the high deformation resistance and springback of titanium alloys by conventional deformation conditions limit the wide application to a certain extent [1,7]. Superplastic forming (SPF) is a promising hot processing technology that enables the formation of complex geometrical parts and reduces the use of fasteners, effectively reducing the weight of structures [8,9]. Moreover, this processing method can improve the utilization rate of materials and cost savings compared to other conventional manufacturing technics [9,10,11].

Superplasticity is the ability of the material to obtain elongation over 200% under certain conditions [12]. The material with stable and fine equiaxed microstructure is prone to obtain excellent superplasticity [13]. For the two-phase titanium alloy, it is easy to obtain a fine grain structure for that the α phase and β phase can be restricted mutually to avoid severe grain growth [14]. The Ti-6Al-4V alloy with optimized volume fraction the β phase effectively improves the superplasticity [15,16]. It is prone to exhibit excellent superplasticity while the volume fraction of the α phase is about 40~50% [15,17]. Moreover, the alloy with fine or ultrafine grain structure can obtain more excellent fracture elongation after superplastic tension than that with coarse or lamellae microstructure [18,19,20,21]. That is, the superplasticity of the alloy depends on various microstructural features like the grain morphology, grain size, and the volume fraction of the β phase [14,22], while the change of the texture can reflect that grain boundary sliding occurs in the alloy during superplastic deformation [23]. Ti55 alloy shows different textures under various tensile strain ratios [24,25]. As the strain increases, the grain rotates, which reduces the intensity of <11$\bar{2}$0> texture, and the proportion of <0001> texture increases [24]. The grain rotation during tensile deformation can coordinate deformation more effectively and increase elongation of the alloy. In the last few years, more and more researchers devoted themselves to investigating the mechanism of superplasticity [26,27,28,29]. The grain boundary sliding accommodated by the dislocation movement and dynamic recrystallization is the main deformation mechanism [27,30,31,32]. By using the in-situ technique, a systematical investigation was carried out that the deformation mechanisms were related to the subgrain size and the average grain size. The results illustrate that the dominant deformation mechanism of superplasticity is grain boundary sliding when the subgrain size is greater than the average grain size. Otherwise, dynamic recrystallization is the predominant deformation mechanism [33].

Over the years, the Ti-6Al-4V alloy is the widely applied superplastic titanium alloy with the optimum SPF temperature of around 900 °C [34]. The relatively high forming temperature is unfavorable to the quality of the product surface and the service life of the die [35]. Ti-6Al-4V alloy with an ultrafine grain microstructure obtained via severe plastic deformation techniques can achieve excellent elongation at a lower temperature of 600 °C [36,37], which limits the wide application. These limitations can be short-came by reformulating the alloy chemistry, such as ATI425 and Ti54M alloy that can obtain a lower optimum forming temperature below 900 °C [38,39,40,41].

SP700 is a new type β-rich α + β titanium alloy with the nominal chemical composition of Ti-4.5Al-3V-2Mo-2Fe (in wt.%) [42]. By adding the β phase stable elements, Fe and Mo, SP700 alloy shows better hot and cold workability, higher strength and toughness, and excellent superplastic behavior at lower temperatures compared with the Ti-6Al-4V alloy [42,43]. With lower SPF temperature, the SP700 alloy can reduce the oxidation of the material and the mold during the forming process. Han et al. investigated the tensile properties and superplasticity of as-forged SP700 alloy with Zr addition [44]. Chen carried out superplastic tensile tests on the SP700 alloy sheet after welding and investigated the microstructure evolution of the weld zone during deformation [45]. Fu et al. studied the superplastic tensile property of SP700 alloy sheet and the mechanical property of superplastic formed conical parts with the superplastic elongation of the sheet samples all over 500% [46,47]. However, there are few studies on the microstructure evolution during superplastic deformation of the SP700 alloy sheet, and in the actual superplastic forming/diffusion bonding (SPF/DB) process, the sheet shows the lowest strain in the diffusion welding point and shows an increasing strain away from the welding point [48]. The amount of strain is of great significance to the superplasticity of the alloy sheet.

The elongation of SP700 alloy with fine grain structure can reach about 2000% at 740~800 °C under constant strain rate, and the optimum SPF temperature is around 760 °C [49], while the strain rate is not constant in the actual superplastic forming process. An optimal process parameter is based on the maximum m value. The m value is the strain rate sensitivity index and represents the necking resistance ability of the alloy during plastic deformation, and specimens can obtain uniform plastic with the higher m value [50]. The maximum m value method is that during the tensile test, the tensile velocity is automatically detected and adjusted so as to maintain the specimen deformed at the optimum strain rate [51]. In our study, superplastic tensile tests on SP700 alloy were also carried out at 740~800 °C by the method of maximum m value. The results showed that SP700 alloy exhibited maximum superplasticity up to 3000% at 760 °C. It is proposed that this should be related to the tensile test method and finer microstructure [46]. As we know, the variation of the microstructure is more obvious with the larger deformation. Therefore, in the present work, the microstructural characteristics at different positions on the sample tested at 760 °C with the elongation of 3000% were evaluated. It is expected to provide a better understanding of the deformation behavior during the SPF process and contribute to expanding applications of this advanced alloy.

## 2. Materials and Experiment

The as-received material was a hot rolled and annealed SP700 alloy (Ti-4.5Al-3V-2Mo-2Fe) sheet with a thickness of 2 mm and a chemical composition (wt.%) shown in Table 1. The β transus temperature was about 910 °C, measured by the metallographic method.

The dimensions of the tensile specimen are shown in Figure 1, with a length of 10 mm and a gauge width of 6 mm. The tensile specimens were machined with the tensile axis parallel to the transverse direction (TD) of the sheet. Superplastic tensile tests at 760 °C were carried out on the SANS-CMT4104 electronic testing machine (MTS, Nanchang, China). Before testing, the surface of the specimen was coated with glass lubricant to prevent oxidation at high temperatures. To get a uniform temperature distribution, 10 min of soaking time was carried out at the given temperatures, and an extensometer was used during tensile tests. In this study, the maximum m-value method was adopted. After tests, the specimens were quenched immediately to keep the high-temperature microstructure.

Similar to the change in strain during SPF/DB, the strain continuously increased from the grip region to the fraction tip position in the superplastic tensile test [30]. The specimens were cut off by wire-electrode cutting for microstructure observation at different locations after superplastic tensile test, as the schematic diagram shown in Figure 2: (i) at the grip position, (ii) at the gauge section near the grip, (iii) at the middle of the gauge section, and (iv) near the tip of the fracture. The specimen surfaces were grounded with a series of SiC abrasive sheets, polished electrochemically with a solution of 5% HCLO_4_ + 95% CH_3_COOH at a voltage of 65 V at room temperature, and etched with Kroll’s reagent (2 mL HF, 8 mL HNO_3_, and 82 mL H_2_O). The microstructure and phase composition of the specimen were studied by the scanning electron microscope (SEM, JEOL JEM-7900, Beijing, China) coupled with the OXFORD Chanel 5 system to perform the electron backscatter diffraction (EBSD) analysis. The pole figures (PFs) and inverse pole figures (IPFs) were calculated from the EBSD data with a step size of 0.05 μm. In this study, when the misorientation angle is less than 15°, the boundaries were definite as low angle boundaries (LAGBs), while the misorientation angle of high angle boundaries (HAGBs) was higher than 15°. To study the morphology of phases and dislocations in the grain of SP700 alloy during superplastic deformation, thin foil samples were prepared by a focused ion beam (FIB) instrument and were evaluated by transmission electron micrograph (TEM) on Tecnai G2 F20. TEM coupled with energy dispersive spectrometry (EDS) was chosen to analyze the distribution of alloy elements.

## 3. Results

### 3.1. Microstructure Evolution

Figure 3 shows the specimen photographs before and after the superplastic tensile test. The samples deformed uniformly without obvious necking during the tensile test. The fracture elongation reached 3000% at 760 °C, indicating that the SP700 alloy exhibited excellent superplasticity.

#### 3.1.1. Initial Microstructure of the SP700 Alloy Sheet

The initial microstructure and texture of the SP700 alloy sheet are shown in Figure 4. The microstructure of the as-received sheet was mainly composed of an equiaxed α phase and a small amount of β phase, which distributed between α particles. There were some slightly elongated grains along the rolling direction. The volume fraction of the α phase was about 87%, while the fraction of the β phase was estimated at only 13%. The average grain size of SP700 alloy was approximately 1.3 μm acquired by EBSD data. The total fraction of LAGBs was 18%, while the fraction of HAGBs was about 82% (Figure 4c). According to the grain orientation spread map (Figure 4d), the majority of the grains had undergone dynamic recrystallization. The pole figure implied the orientation of the initial sheet grains is demonstrated in Figure 4e. From the (0001) pole figure, the c-axis directions were approximately parallel to the normal direction (ND), with an angle of about 16°. The texture component of the α phase could be represented as (10$\bar{1}$6)<1$\bar{1}$00> with a maximum pole density of 11.2.

#### 3.1.2. Microstructure Evolution during Superplastic Deformation

The strain continuously increased from the grip region to the fraction tip position in the superplastic tensile test. Microstructures at various positions of the sample deformed at 760 °C were observed to study the microstructure evolution during deformation. The SEM images and phase fraction maps were illustrated in Figure 5, and the average grain size, as well as phase ratio, were statistics in Figure 6. It was noteworthy that the grip region (position ⅰ) only experienced statical annealing without any deformation during the superplastic tensile test. The average grain size in the grip region was about 2.5 μm. In the gauge section, it could be seen that the grains had grown up obviously from the near grip section to the tip section. As shown in Figure 5a–d, the average grain size of α and β phase in the near tip position could reach 4.7 μm and 5.5 μm, respectively. Additionally, the β volume fraction increased as the strain increased. According to Figure 6, the β volume fraction in the grip section was about 30%, whereas it increased to 40% at the tip section.

Misorientation angle distribution and the grain orientation spread (GOS) maps were calculated at different tensile test positions, as shown in Figure 7. Compared with the initial microstructure in Figure 4c,d, the proportion of LAGBs decreased at the grip region of the sample, which only experienced statical annealing, and the average misorientation angle changed from 32.5° to 33.2°; that is, static recrystallization occurred in the grip section. With the increase in strain, the fraction of LAGBs decreased first and then rose, and the average grain misorientation angles were 33.2°, 36.2°, 35.3°, and 32.6°, while the low GOS value (usually lower than 1°, represented by blue color) was related to dynamic recovery or dynamic recrystallization [52]. As shown in Figure 7h, some highly deformed grains existed with a high GOS value with the elongation of 3000%.

### 3.2. Texture Evolution during Superplastic Deformation

To better understand the changes of texture during deformation, the IPFs of the SP700 sample at various positions are shown in Figure 8. Compared with the texture of the initial microstructure (shown in Figure 4e), there was no obvious change of that at the grip position, as shown in Figure 8a, which was dominated by texture (0001) with an intensity of 3.3. The grip section only underwent a further annealing process, resulting in a less significant change to textures. As the strain increased, the initial texture (0001) weakened, and the textures changed from (0001) to (10$\bar{1}$0), as shown in Figure 8b–d. That meant more and more grain rotated with the basal plane perpendicular to the RD-TD plane, and the c-axis of the α grain is parallel to the tensile test direction (TD direction). In the near tip section, the intensity of (0001) texture further was only about 1, while that of (10$\bar{1}$0) texture rose to 1.7.

## 4. Discussion

### 4.1. Microstructure Aspects of Superplasticity

In this study, an excellent superplasticity of 3000% at 760 °C was obtained, compared with the previously reported results [44]. It is well known that grain size and phase ratio are two of the most important factors for the superplasticity of the material [15]. As shown in Figure 4a, the average grain size of this alloy was ~1.3 μm, which was much smaller than the requirement of 10 μm for superplasticity. Compared with the Ti-6Al-4V alloy, which showed superplasticity of about 1300% at 900 °C and the grain size of this alloy was ~2.5 μm [30], the finer grain size and 30~40% content of the β phase were attributed to the outstanding superplasticity.

The strain continuously increased from the grip region to the fraction tip position in the superplastic tensile test. For two-phase titanium alloy, the diffusion rate of the β phase was two orders of magnitude higher than that of the α phase at the same temperature [53]. From Figure 6, the β phase grain grew faster than the α phase with increasing strain, while the temperature, time, and strain rate of all deformed regions were the same during tension, namely, stress accelerated grain growth which was unfavorable to the superplastic elongation of the alloy. It was also observed that the volume fraction of the β phase increased as the strain increased (Figure 6); one reason may be that the α→β phase transformation occurred during deformation.

It is well known that concentrations of alloying elements are dissimilar within the α and β phases. The phase compositions of the grip and near the tip section were analyzed by TEM-EDS, and the results were illustrated in Table 2. It is noted that the Fe element, as a strong β-stabilizing and fast diffusing element, was almost all dissolved in the β phase. The contents of Mo and V elements in the β phase were much more than those in the α phase for the undeformed specimen, and the differences were further increased after deformation. The volume fraction of the β phase is also increased, that is, the segregation of Mo, Fe, and V elements within the β phase is accelerated during deformation. As reported, Fe and Mo elements diffused faster than Al element in titanium [54,55]. The tensile stress promotes the diffusion of β stabilizers from α to β phase, resulting in the α→β phase transformation.

According to the previous studies, the mechanism of superplastic deformation mainly included dislocation slip [56], grain boundary sliding, and diffusion creep [28,29,33]. In the α + β titanium alloy, there were three types of boundaries, namely α phase boundaries (α/α), β phase boundaries (β/β), and interfaces between α and β phases (α/β). The sliding resistance of different boundaries increased in the order of α/β ≪ α/α ≈ β/β [31,33]. Therefore, grain boundary sliding mainly occurred at the interfaces between α and β phases (α/β). Since the active slip systems of β phase with body center cubic (bcc) structure were more than that of α phase with hexagonal close-packed crystal (hcp) structure, the β phase could be regarded as the “soft phase” and suffers larger plastic deformation [57]. Therefore, the high-volume fraction of the β phase was desired. However, the grain would grow rapidly, and the phase boundary fraction decreased with a high-volume fraction of the β phase [58] so that an optimal β phase volume fraction was beneficial to improve superplasticity [59].

Figure 6 showed that the grain size and the volume fraction of the β phase increased as the strain increased. The volume fraction of α/β phase boundary at different positions was calculated, as illustrated in Figure 9. The volume fraction of α/β phase boundary showed 58.9% at the grip section (ⅰ), while it rose up to 61.6% near the grip section (ⅱ). The increase in the volume fraction of α/β phase boundary was beneficial to uniform deformation and could avoid necking at the initial stage of deformation. As the strain increased, the volume fraction of the α/β phase boundary decreased and reduced to 53.5% at the tip section (ⅳ). In this study, the volume fraction of the β phase ranged from 30% to 40%. With relatively uniform content of β phase, the sample exhibited uniform deformation without obvious necking and showed excellent elongation with 3000% at 760 °C.

### 4.2. Texture Aspects of Superplasticity

The results presented above suggested an excellent deformation response for the SP700 alloy under superplastic conditions. In the superplastic deformation process, the texture is weakened due to the spread around texture components. This was mainly due to the rotation of grain caused by boundary sliding [22]. At the beginning of the deformation, the (0001) plane of the α phase was parallel to the RD-TD plane of the sheet. During deformation, the texture split along with TD in the basal pole figures in the gauge section, as shown in Figure 10 by the arrow. The grain boundary sliding made the grain lattice rotate slightly under stress, and the (0001) plane was still approximately parallel to the RD-TD plane. Figure 11 shows the unit cell orientation of HCP crystal at various deformation positions at 760 °C. Under the external force, the c-axis of the crystal lattice rotated, resulting in the (10$\bar{1}$0) texture at the tip region, which was unfavorable for the slip system to glide. Hence, the increase in strain led to the accumulation of dislocations, which resulted in the increase in the proportion of LAGBs, as illustrated in Table 3.

Figure 12 shows the TEM morphologies of α and β phases at the grip section and the gauge section after superplastic tension at 760 °C. The α/β boundaries were curved and dislocations arrayed on both sides of the grain boundary, indicating that phase boundary sliding accommodated by dislocation slip and diffusion was the main deformation mechanism of SP700 alloy during superplastic deformation. However, due to practical constraints, this paper cannot provide a comprehensive review of the direct influence of the α/β boundaries on superplastic elongation and the deformation mechanism of SP700 alloy. In addition, the effect of temperature was also unclear. These will be further studied in later research.

## 5. Conclusions

(1)The SP700 alloy sheet with a grain size of 1.3 μm showed excellent superplasticity at 760 °C with the fraction elongation up to 3000% using the maximum m value method.(2)During the superplastic deformation process, the microstructure kept fine equiaxed grains with the grain size increasing from 2.5 μm to 5.5 μm as the strain increased. Meanwhile, the β phase volume fraction increased from 30% to 40% due to the diffusion of Al, Mo, Fe, and V elements with the higher value of [Mo]_eq_ in the β phase.(3)During the deformation process, the intensity of the texture and the dominant texture changed as the deformation strain increased, indicating that the grain rotation occurred. The grain boundary sliding accommodated by the grain rotation and dislocation slip was the main deformation mechanism of SP700 alloy.

## Figures and Tables

**Figure 1 materials-15-01808-f001:**
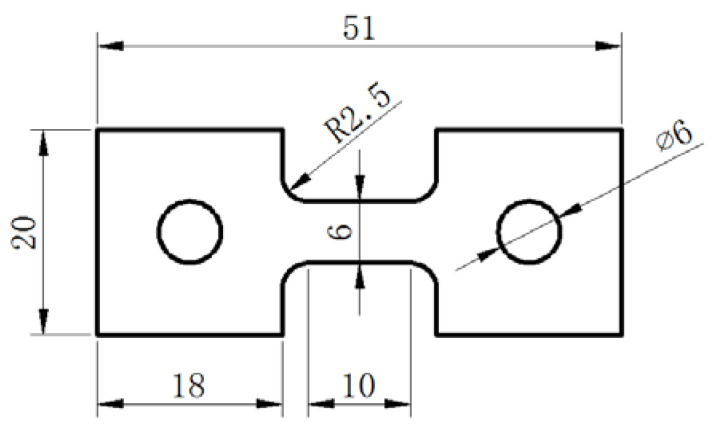
The dimensions of the tensile specimen used for the superplastic test (mm).

**Figure 2 materials-15-01808-f002:**
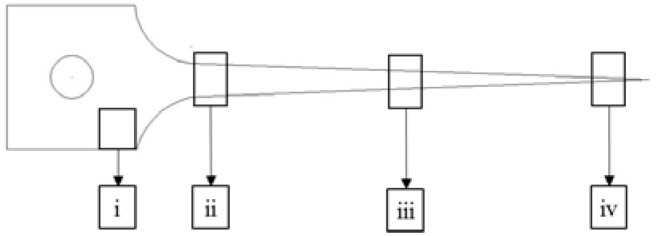
The schematic diagram of the sample locations cut from the tensile test samples: (i) at the grip position, (ii) at the gauge section near the grip, (iii) at the middle of the gauge section, and (iv) near the tip of the fracture.

**Figure 3 materials-15-01808-f003:**
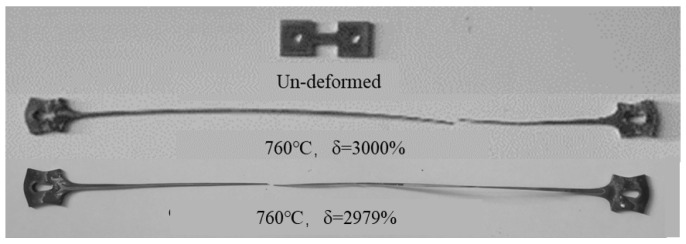
The samples photo before and after superplastic tensile test.

**Figure 4 materials-15-01808-f004:**
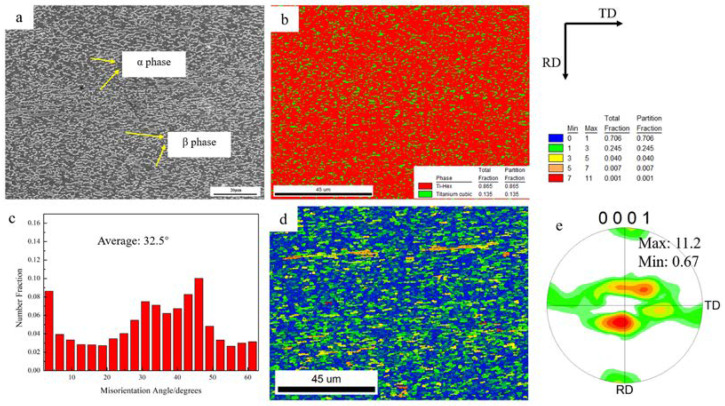
Microstructure and texture of the initial sheet: (**a**) SEM image, (**b**) phase fraction map, (**c**) misorientation angle chart, (**d**) grain orientation spread map, and (**e**) (0001) pole figure of α phase.

**Figure 5 materials-15-01808-f005:**
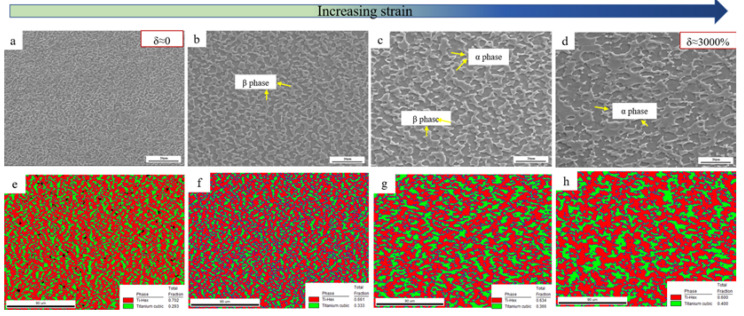
SEM images and phase fraction maps of SP700 alloy at different positions at 760 °C: (**a**,**e**) the grip position, (**b**,**f**) the gauge section near the grip, (**c**,**g**) the middle of the gauge section, and (**d**,**h**) section near the tip of the fracture.

**Figure 6 materials-15-01808-f006:**
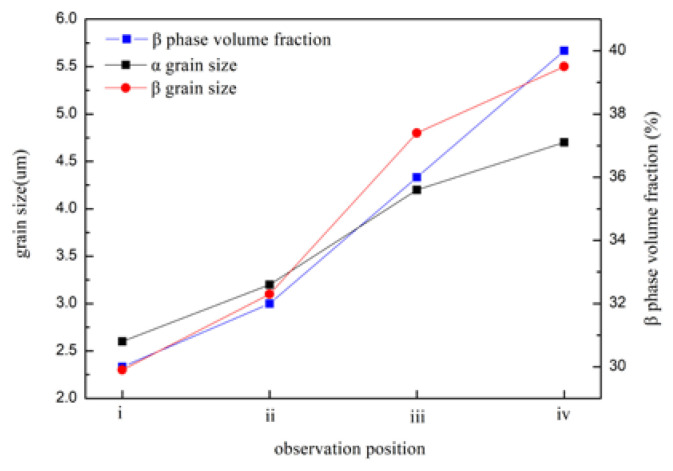
The phase grain sizes and β phase volume fraction at various tensile test positions.

**Figure 7 materials-15-01808-f007:**
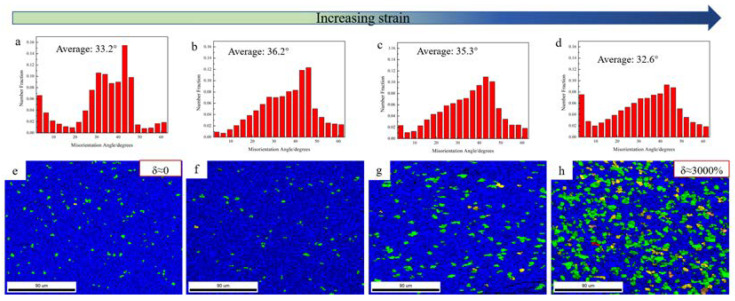
Misorientation angle charts and grain orientation spread maps of SP700 alloy at different positions at 760 °C: (**a**,**e**) the grip position, (**b**,**f**) the gauge section near the grip, (**c**,**g**) the middle of the gauge section, and (**d**,**h**) section near the tip of the fracture.

**Figure 8 materials-15-01808-f008:**
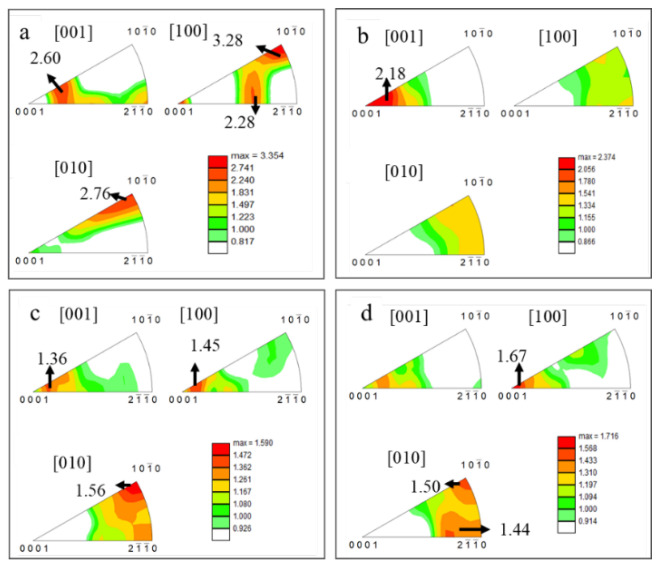
IPFs at various deformation positions at 760 °C: (**a**) at the grip position, (**b**) the gauge section near the grip, (**c**) the middle of the gauge section, and (**d**) section near the tip of the fracture.

**Figure 9 materials-15-01808-f009:**
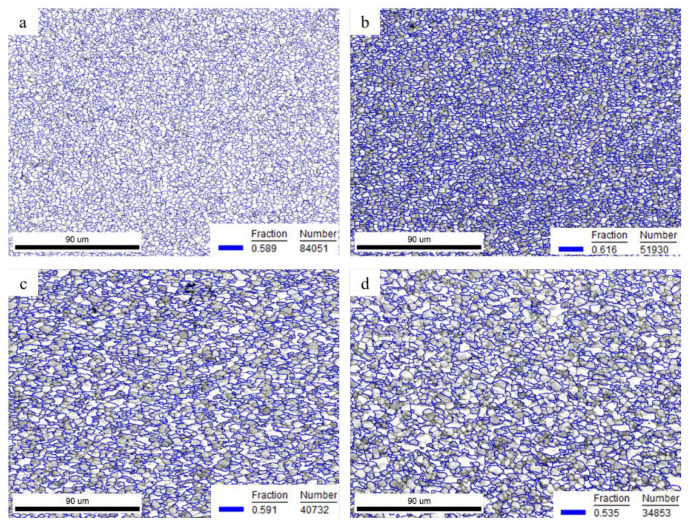
The spread of α/β phase boundaries at different positions deformed at 760 °C: (**a**) the grip position, (**b**) the gauge section near the grip, (**c**) the middle of the gauge section, and (**d**) section near the tip of the fracture.

**Figure 10 materials-15-01808-f010:**
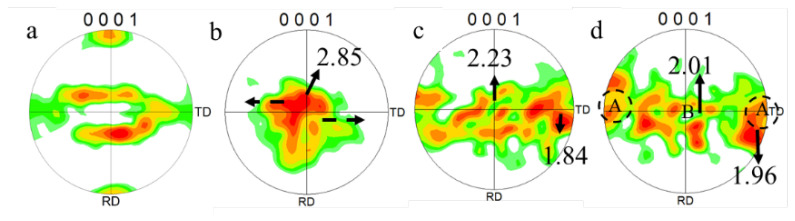
Pole figures (PFs) at various deformation positions at 760 °C: (**a**) at the grip position, (**b**) the gauge section near the grip, (**c**) the middle of the gauge section, and (**d**) section near the tip of the fracture.

**Figure 11 materials-15-01808-f011:**
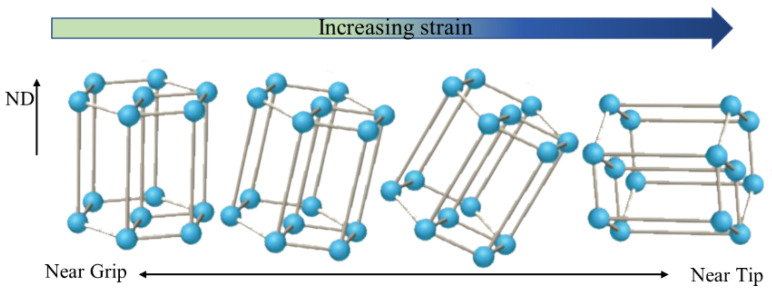
Unit cell orientation of HCP crystal at various deformation positions at 760 °C.

**Figure 12 materials-15-01808-f012:**
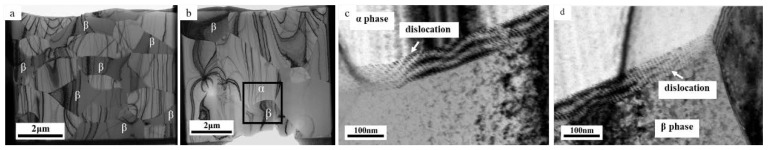
Transmission electron micrographs of specimen deformed at 760 °C: (**a**) at the grip position and (**b**–**d**) at the middle of the gauge section.

**Table 1 materials-15-01808-t001:** The chemical composition of SP700 titanium alloy (wt.%).

Element	Ti	Al	V	Fe	Mo	O	H
wt.%	Bal.	4.36	3.10	1.94	2.00	0.12	<0.001

**Table 2 materials-15-01808-t002:** The chemical compositions of α and β phases at different positions after tension at 760 °C (wt.%).

Position	Phase	Ti	Al	V	Fe	Mo	[Mo]_eq_
Grip	β	83.53	3.93	4.29	3.20	5.05	17.20
	α	92.50	5.30	1.27	0.09	0.84	1.95
Tip	β	81.03	2.59	5.47	4.79	6.12	23.67
	α	92.67	5.69	0.95	0.08	0.60	1.46

**Table 3 materials-15-01808-t003:** The fraction of LAGBs at different positions in the tensile test specimen.

Position in the Specimen	Before Test	Grip	Near Grip	Middle of the Gauge	Tip
Fraction of LAGBs/%	21	7	5	9	15

## Data Availability

The data presented in this study are available on request from the corresponding author.

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
