# Peer review of "Microstructure and Texture Evolution during Superplastic Deformation of SP700 Titanium Alloy"

_materials, 2022, doi:10.3390/ma15051808_

Round 1
Reviewer 1 Report
The authors have presented an interesting manuscript on the microstructure and texture evolution of SP700 titanium alloy during its superplastic deformation. Although the microstructural investigations were performed correctly and results were presented in a good manner, they are still based on only one tensile test carried out at one specific condition. One cannot rely on one test only especially when the superplastic deformation of SP700 was already studied in the literature. It should be also highlighted, that uncertainties resulting from high-temperature uniaxial tensile testing could also affect the results significantly. The authors stated, that uniform temperature distribution was ensured by 10 minutes of soaking time. It was not mentioned if the temperature of the specimen was monitored anyhow. The different microstructure from different areas of the specimen may result from higher or lower temperatures than 760C. Also, the temperature selection and testing conditions were not justified. The reviewer appreciates the microstructural characterization and detailed analysis performed by the authors however, the manuscript can not be accepted in a presented way.
Author Response
Dear Professor,
We sincerely thank you very much for the valuable feedback that we have used to improve the quality of our manuscript. The reviewer comments are laid out below in italicized font and specific concernshave been numbered. Our response is given in normal font and changes/additionsto the manuscript are given in red text.
Comment: The authors have presented an interesting manuscript on the microstructure and texture evolution of SP700 titanium alloy during its superplastic deformation. Although the microstructural investigations were performed correctly and results were presented in a good manner, they are still based on only one tensile test carried out at one specific condition. One cannot rely on one test only especially when the superplastic deformation of SP700 was already studied in the literature. It should be also highlighted, that uncertainties resulting from high-temperature uniaxial tensile testing could also affect the results significantly. The authors stated, that uniform temperature distribution was ensured by 10 minutes of soaking time. It was not mentioned if the temperature of the specimen was monitored anyhow. The different microstructure from different areas of the specimen may result from higher or lower temperatures than 760C. Also, the temperature selection and testing conditions were not justified. The reviewer appreciates the microstructural characterization and detailed analysis performed by the authors however, the manuscript can not be accepted in a presented way.
Response: According to your valuable opinions, we summarize that the selection of experiment temperature and the uniformity of the temperature during tensile test. During the tensile test, we used the temperature detector to ensure the accuracy and uniform temperature distribution. And 10 minutes of soaking time is enough for the samples used in this paper based on multiple tensile tests we have done.
We added to page 3 (Paragraph 5, Section â…°):
The elongation of SP700 alloy with fine grain structure can reach about 2000% at 740~800℃ under constant strain rate. While the strain rate is not constant in the actual superplastic forming process. An optimal process parameter for strain rate corresponds to the maximum m value. The m value is the strain rate sensitivity index and represents the necking resistance ability of the alloy during plastic deformation, and specimens can obtain uniform plastic with the higher m value. The maximum m value method is that during the tensile test, the tensile velocity is automatically detected and adjusted so as to maintain the specimen deformed at the optimum strain rate. In our study, superplastic tensile tests on SP700 alloy were also carried out at 740℃~800℃ by the method of maximum m value. The results showed that SP700 alloy exhibited maximum superplasticity up to 3000% at 760℃. It is proposed that this should be related to the tensile test method and finer microstructure. As we know, the variation of the microstructure is more obvious with the lager deformation. Therefore, in the present work, the microstructural characteristics at different positions on the sample tested at 760℃ with the elongation of 3000% were evaluated.
Special thanks to you for your good comments. We tried our best to improve the manuscript and made some changes in the manuscript. These changes will not influence the content and framework of the paper. We appreciate for Reviewers’ warm work earnestly, and hope that the correction will meet with approval. Once again, thank you very much for your comments and suggestions.
With best wishes,
Yours sincerely,
Wenjun Ye and Xiaoyun Song.

Reviewer 2 Report
Based on the results of an in-depth evaluation that I have done for an article with the title “Microstructure and texture evolution during superplastic deformation of SP700 titanium alloy” by Tian et al. (Special issue in Quality, Microstructure and Properties of Metal Alloys), I think this article should be rejected for publication in Materials or reconsider after proper changes in major revision.
- I would encourage and advise the authors to adopt some of the additional references in the introduction section published by MDPI as follow:
- Tresca Stress Simulation of Metal-on-Metal Total Hip Arthroplasty during Normal Walking Activity. Materials (Basel). 2021, 14, 7554. https://doi.org/10.3390/ma14247554
- The Effect of Bottom Profile Dimples on the Femoral Head on Wear in Metal-on-Metal Total Hip Arthroplasty. J. Funct. Biomater. 2021, 12, 38. https://doi.org/10.3390/jfb12020038
- I see the inappropriate use of English in some areas of the present manuscript. To improve the quality of English used in this manuscript and make sure English language, grammar, punctuation, spelling, and overall style are correct, further proofreading is needed. As an alternative, the authors can use the MDPI English proofreading service for this issue.
- The author seems to have made an error in using uppercase and lowercase in the title (line 2-3) of the present article that should be corrected.
- Describe the novelty of the article made by the author? From the results of my evaluation, it seems that many similar published works adequately explain what you have raised in the current manuscript. Superplasticity has been widely investigated in some materials. However, only evaluating the superelasticity phenomena on SP700 alloy at 760℃ is not enough novelty to raisin in the author’s present article. If there are something others really new in this manuscript, please highlight it more clearly in the introduction.
- In the materials and methods section (line 85-133) the authors should add one systematic figure to illustrate the workflow of experimental testing in the present study to make the reader more interested and easier to understand rather than only using dominant text to explain.
- The author must provide a detailed specification and use condition more detail regarding all tools used in the research carried out so that the reader can estimate the accuracy and differences in the results that the authors describe due to the use of different tools in future studies. For example X-ray diffractometer (XRD, Rigaku D/Max Ultima III, Tokyo, Japan) with specification ……, use condition ……
- The authors should arrange the figure location, please make sure the figure and description figure is on the same page, such as Figure 2 The schematic diagram of the sample locations cut from the tensile….. in line 128-129. The figure is in Page 3, but the description figure is in page 4.
- In the result and discussion section (line 132-287), if possible, the authors should compare the results in the present study with the published literature in the similar/identical condition and explain this comparison become comprehensive discussion to ensure the result obtained in the present study.
- In the discussion section (line 204-287), the authors should add at least one paragraph to describe the limitation of the research carried out that would be added in the next of line 287.
- The conclusion (line 288-305) of the present manuscript is not solid. Further elaboration is needed. Also, make it into paragraphs, not point-by-point as presented in the author’s manuscript.
- Further research needs to be explained in the conclusion section (line 288-305).
- Please make sure the author has used the Materials, MDPI format correctly because there are some errors found in this manuscript regarding the format, such as typesetting on authors contribution until references (line 308-476). The authors can download published manuscripts by Materials, MDPI, and compare them with the present author's manuscript to ensure typesetting is appropriate.
Author Response
Dear Professor,
We sincerely thank you very much for the valuable feedback that we have used to improve the quality of our manuscript. The reviewer comments are laid out below in italicized font and specific concernshave been numbered. Our response is given in normal font and changes/additionsto the manuscript are given in blue text.
Comment 1: I would encourage and advise the authors to adopt some of the additional references in the introduction section published by MDPI as follow:
- Tresca Stress Simulation of Metal-on-Metal Total Hip Arthroplasty during Normal Walking Activity. Materials (Basel). 2021, 14, 7554. https://doi.org/10.3390/ma14247554
- The Effect of Bottom Profile Dimples on the Femoral Head on Wear in Metal-on-Metal Total Hip Arthroplasty. J. Funct. Biomater. 2021, 12, 38. https://doi.org/10.3390/jfb12020038
Response: As suggested by the reviewer, we have added more references into the INTRODUCTION part in the revised manuscript (referred to Paragraph 1, Ref.5 and Ref. 6).
Ref.5: Basri, H. The Effect of Bottom Profile Dimples on the Femoral Head on Wear in Metal-on-Metal Total Hip Arthroplasty. Journal of Functional Biomaterials. 2021, 12.doi:https://doi.org/10.3390/jfb12020038.
Ref.5: Jamari, J. Tresca Stress Simulation of Metal-on-Metal Total Hip Arthroplasty during Normal Walking Activity. Materials. 2021, 14.doi: https://doi.org/10.3390/ma14247554.
Comment 2: I see the inappropriate use of English in some areas of the present manuscript. To improve the quality of English used in this manuscript and make sure English language, grammar, punctuation, spelling, and overall style are correct, further proofreading is needed. As an alternative, the authors can use the MDPI English proofreading service for this issue.
Response: We feel sorry for our poor writings. We tried our best to improve the manuscript and made some changes in the manuscript. These changes will not influence the content and framework of the paper.
Comment 3: The author seems to have made an error in using uppercase and lowercase in the title (line 2-3) of the present article that should be corrected.
Response: We feel sorry for our carelessness. In our resubmitted manuscript, we have corrected the “Microstructure and texture evolution during superplastic deformation of SP700 titanium alloy” to “Microstructure and Texture Evolution during Superplastic Deformation of SP700 Titanium Alloy”. Thanks for your correction.
Comment 4: Describe the novelty of the article made by the author? From the results of my evaluation, it seems that many similar published works adequately explain what you have raised in the current manuscript. Superplasticity has been widely investigated in some materials. However, only evaluating the superelasticity phenomena on SP700 alloy at 760℃ is not enough novelty to raisin in the author’s present article. If there are something others really new in this manuscript, please highlight it more clearly in the introduction.
Response: In order to explain the selection of experiment temperature we added some information to page 3 (Paragraph 5, Section â…°):
The elongation of SP700 alloy with fine grain structure can reach about 2000% at 740~800℃ under constant strain rate. While the strain rate is not constant in the actual superplastic forming process. An optimal process parameter for strain rate corresponds to the maximum m value. The m value is the strain rate sensitivity index and represents the necking resistance ability of the alloy during plastic deformation, and specimens can obtain uniform plastic with the higher m value. The maximum m value method is that during the tensile test, the tensile velocity is automatically detected and adjusted so as to maintain the specimen deformed at the optimum strain rate. In our study, superplastic tensile tests on SP700 alloy were also carried out at 740℃~800℃ by the method of maximum m value. The results showed that SP700 alloy exhibited maximum superplasticity up to 3000% at 760℃. It is proposed that this should be related to the tensile test method and finer microstructure. As we know, the variation of the microstructure is more obvious with the lager deformation. Therefore, in the present work, the microstructural characteristics at different positions on the sample tested at 760℃ with the elongation of 3000% were evaluated.
Comment 5: In the materials and methods section (line 85-133) the authors should add one systematic figure to illustrate the workflow of experimental testing in the present study to make the reader more interested and easier to understand rather than only using dominant text to explain.
Response: Thank you for your positive comments. While the main purpose in this study is to analyze the microstructure evolution at different deformation positions of the sample after superplastic tensile test and the schematic diagram of the sample location was shown in Figure 2 in the text. There are no other complex experimental tests, so it seems not necessary to draw the systematic figure in this paper. However, your valuable suggestions are beneficial to the subsequent work for me.
Comment 6: The author must provide a detailed specification and use condition more detail regarding all tools used in the research carried out so that the reader can estimate the accuracy and differences in the results that the authors describe due to the use of different tools in future studies. For example X-ray diffractometer (XRD, Rigaku D/Max Ultima III, Tokyo, Japan) with specification ……, use condition ……
Response: Thanks again and the detailed specification and use condition have been carefully supplemented in the MATERIALS AND EXPERIMENT part accordingly(referred to Paragraph 3, Section â…±).
Comment 7: The authors should arrange the figure location, please make sure the figure and description figure is on the same page, such as Figure 2 The schematic diagram of the sample locations cut from the tensile….. in line 128-129. The figure is in Page 3, but the description figure is in page 4.
Response: Sorry for our carelessness. We have carefully checked the manuscript and adjusted the layout of the article to avoid the inaccurate arrangement of the figure and description figure locations.
Comment 8: In the result and discussion section (line 132-287), if possible, the authors should compare the results in the present study with the published literature in the similar/identical condition and explain this comparison become comprehensive discussion to ensure the result obtained in the present study.
Response: Thank you for your positive comments and we have added the comparation of mechanical properties between Ti-6Al-4V and SP700 alloy (Paragraph 1, Section â…³):
Compared with Ti-6Al-4V alloy which showed superplasticity of about 1300% at 900℃ and the grain size of this alloy was ~2.5μm, the finer grain size and 30%~40% content of the β phase were attributed to the outstanding superplasticity.
Comment 9: In the discussion section (line 204-287), the authors should add at least one paragraph to describe the limitation of the research carried out that would be added in the next of line 287.
Response: Thank you for your reminding and we have added one paragraph which describes the limitation of the research (Paragraph 7, Section â…³):
However, due to practical constraints, this paper cannot provide a comprehensive review of the direct influence of the α/β boundaries on superplastic elongation and the deformation mechanism of SP700 alloy. In addition, the effect of temperature was also unclear. These will be further studied in later research.
Comment 10: The conclusion (line 288-305) of the present manuscript is not solid. Further elaboration is needed. Also, make it into paragraphs, not point-by-point as presented in the author’s manuscript.
Response: Thank you for your reminding and we have corrected these mistakes based on your suggestions:
5.Conclusion
(1) The SP700 alloy sheet with grain size of 1.3μm showed excellent superplasticity at 760℃ with the fraction elongation up to 3000% using the maximum m value method.
(2) During the superplastic deformation process, the microstructure kept fine equiaxed grains with the grain size increasing from 2.5μm to 5.5μm as the strain increased. Meanwhile the β phase volume fraction increased from 30% to 40% due to the diffusion of Al, Mo, Fe and V elements with the higher value of [Mo]eq in β phase.
(3) During the deformation process, the intensity of the texture and the dominant texture changed as the deformation strain increased, indicating that the grain rotation occurred. The grain boundary sliding accommodated by the grain rotation and dislocation slip was the main deformation mechanism of SP700 alloy.
Comment 11: Further research needs to be explained in the conclusion section (line 288-305).
Response: We agree that more study would be useful to understand details of the superplastic deformation mechanism of SP700 titanium alloy. We hope, in the future, to employ in-situ techniques to better explain the function of dislocation and α/β phase boundaries. The XRD techniques is also expected to be applied to help the correlation with the analysis.
Comment 12: Please make sure the author has used the Materials, MDPI format correctly because there are some errors found in this manuscript regarding the format, such as typesetting on authors contribution until references (line 308-476). The authors can download published manuscripts by Materials, MDPI, and compare them with the present author's manuscript to ensure typesetting is appropriate.
Response: We were really sorry for the careless mistakes. Thank you for your reminding and we have corrected these mistakes based on your suggestions.
Special thanks to you for your good comments. We tried our best to improve the manuscript and made some changes in the manuscript. These changes will not influence the content and framework of the paper. We appreciate for Reviewers’ warm work earnestly, and hope that the correction will meet with approval. Once again, thank you very much for your comments and suggestions.
With best wishes,
Yours sincerely,
Wenjun Ye and Xiaoyun Song.

Reviewer 3 Report
The paper ”Microstructure and texture evolution during superplastic deformation of SP700 titanium alloy” is suitable for publication in Materials Journal just with minor updates.
- in the introduction please update some specific results with values (extra references) of mechanical properties and texture for Ti-6Al-4V and SP700
- parameters of the microstructure characteristics are missing
- XRD analysis is needed in order to observe the crystalline structure with the lattice parameters; this will help the correlation with the texture analysis.
Author Response
Dear Professor,
We sincerely thank you very much for the valuable feedback that we have used to improve the quality of our manuscript. The reviewer comments are laid out below in italicized font and specific concernshave been numbered. Our response is given in normal font and changes/additionsto the manuscript are given in green text.
Comment 1: in the introduction please update some specific results with values (extra references) of mechanical properties and texture for Ti-6Al-4V and SP700
Response: Thank you for your positive comments and we have added the comparation of mechanical properties between Ti-6Al-4V and SP700 alloy (Paragraph 1, Section â…³):
Compared with Ti-6Al-4V alloy which showed superplasticity of about 1300% at 900℃ and the grain size of this alloy was ~2.5μm, the finer grain size and 30%~40% content of the β phase were attributed to the outstanding superplasticity.
Comment 2: parameters of the microstructure characteristics are missing
Response: Thank you for your positive comments. We hope to use XRD techniques to analysis the parameters of the microstructure characteristics to understand details of interaction and enhancement in the future.
Comment 3: XRD analysis is needed in order to observe the crystalline structure with the lattice parameters; this will help the correlation with the texture analysis.
Response: In this experiment, the SP700 alloy tested shows uniform microstructure and the observation area by EBSD technique is approximate to 1mm2. These ensure the accuracy of the data to some extent. We agree that more study would be useful to understand details of interaction and enhancement. We hope, in the future, to employ XRD analysis to assist the texture analysis.
Special thanks to you for your good comments. We tried our best to improve the manuscript and made some changes in the manuscript. These changes will not influence the content and framework of the paper. We appreciate for Reviewers’ warm work earnestly, and hope that the correction will meet with approval. Once again, thank you very much for your comments and suggestions.
With best wishes,
Yours sincerely,
Wenjun Ye and Xiaoyun Song.

Round 2
Reviewer 1 Report
Since the authors stated that superplastic tensile tests on SP700 alloy were also carried out at 740℃~800℃ by the method of maximum m value, microstructural observations and analyses should be provided for different cases as well. One cannot deduce from one test only.
Author Response
Dear Professor,
Firstly, we would like to thank you for your kind letter and constructive comments concerning our article. These comments are all valuable and helpful for improving our article. All the authors have seriously discussed about all these comments. According to the reviewers’ comments, we have tried best to modify our manuscript to meet with the requirements. In this revised version, changes to our manuscript within the document were all highlighted by using red colored text. Point-by-point responses to the reviewers are listed below this letter.
Comment: Since the authors stated that superplastic tensile tests on SP700 alloy were also carried out at 740℃~800℃ by the method of maximum m value, microstructural observations and analyses should be provided for different cases as well. One cannot deduce from one test only.
Response: The main purpose of our study is expected to provide a better understanding of the deformation behavior during the superplastic forming (SPF) process and contribute to expanding applications of this advanced alloy. The optimum SPF temperature of SP00 alloy is around 760 °C[Ref.49]. In our study, the SP700 alloy exhibited maximum superplasticity up to 3000% at 760℃ by the method of maximum m value. Therefore, the main research object is the sample tested at 760°C. And the temperature monitor was used to ensure the uniformity of the temperature during the tensile test process. At the same time, tensile tests were carried out on two parallel samples to avoid the contingency.
We agree that more study would be useful to understand details of interaction and enhancement. However, the effect of temperature on superplasticity of the alloy is the main content of the follow-up research. At present, through preliminary exploration, the microstructure and texture evolution of the SP700 tested at the temperature range of 740°C~800°C is similar to that at 760°C.
Special thanks to you for your good comments. We tried our best to improve the manuscript and made some changes in the manuscript. These changes will not influence the content and framework of the paper. If there are any other modifications we could make, we would like very much to modify them and we really appreciate your help. Once again, thank you very much for your comments and suggestions.
With best wishes,
Yours sincerely,
Wenjun Ye and Xiaoyun Song.

Reviewer 2 Report
Dear Tian et al.,
After carefully reading the author's revised manuscript entitled "Microstructure and texture evolution during superplastic deformation of SP700 titanium alloy" (materials-1596535) by Tian et al., The authors have been made significant improvements in the revised manuscript. Also, all of the issue in my review report have been addressed precisely.
With my pleasure, I recommend the manuscript should be accepted for publication on Materials.
Best regards,
The Reviewer
Author Response
Dear Professor,
Firstly, we would like to thank you for your kind letter and constructive comments concerning our article. These comments are all valuable and helpful for improving our article. All the authors have seriously discussed about all these comments.
Special thanks to you for your good comments. We have tried our best to polish the language in the revised manuscript. These changes will not influence the content and framework of the paper.
If there are any other modifications we could make, we would like very much to modify them and we really appreciate your help. Once again, thank you very much for your comments and suggestions.
With best wishes,
Yours sincerely,
Wenjun Ye and Xiaoyun Song.
Round 3
Reviewer 1 Report
The reviewer understands the author's approach. The final decision, however, will be made by Special Issue Editor.